# Real-Time Environmental Monitoring Platform for Wellness and Preventive Care in a Smart and Sustainable City with an Urban Landscape Perspective: The Case of Developing Countries

Victor Gonzalez *, Manuel Peralta, Juan Faxas-Guzmán and Yokasta García Frómeta

Pontificia Universidad Católica Madre y Maestra, Santo Domingo 10114, Dominican Republic
* Correspondence: vm.gonzalez@ce.pucmm.edu.do; Tel.: +1-809-654-0534

**Abstract:** Smart and sustainable communities seek to ensure comfortable and sustainable quality of life for community residents, the environment and the landscape. Pollution is a key factor affecting quality of life within a community. This research provides a detailed insight into a successfully developed and deployed framework for an environmental monitoring platform for an urban study to monitor, in real time, the air quality and noise level of two cities of the Dominican Republic—Santo Domingo and Santiago de Los Caballeros. This urban platform is based on a technology range, allowing for the integration of multiple environmental variables related to landscape and providing open data access to urban study and the community. Two case studies are presented: The first highlights how the platform can be used to understand the impact a natural event, for example, how dust landscapes (such as the Sahara) impact a community and the actions that can be taken for wellness and preventive care. The second case focuses on understanding how policies taken to prevent the spread of COVID-19 affect the air quality and noise level of the landscape and community. In the second case, the platform can be used to expand the view of decision makers in the urban landscape and communities that are affected.

**Keywords:** smart and sustainable city; landscape; urban studies; internet of things; air quality and noise monitoring; wellness and preventive care; COVID-19





## 1. Introduction

Smart and sustainable cities are an expression of the multiple domains of urban life in which technology and policies can be applied to a community within their landscape [1]. The vision has progressed to focus on the alignment between policies related to human capital, education, economic development, territorial and governance, and how these could be improved with the use of ICT [2]. To become a smart and sustainable community, its development must combine ICT with territorial, human and social capital, along with broad economic policies [3]. Currently, the development of smart cities seeks to ensure a comfortable and sustainable quality of life for the residents of the community, the environment and their landscape [4]. This creates a bigger emphasis on the sustainability of the community and the surrounding territories [5].

One of the biggest threats to sustainability for both a territory and a community is the level of pollution that surrounds them. Urban communities tend to be affected by low air quality and high noise levels [6]. These pollutants have a direct effect on the wellbeing of a community—there is evidence that links them to an increase in deaths, admissions to hospitals, and general problems in the respiratory and immune systems of residents [7,8]. In addition, these contaminants have an impact on the territory as urbanization changes the land surface temperature, and this is critical for the decision-making process of sustainable and vibrant city development [9]. Scholars have recently focused on understanding the

behavior of the pollutants as a preventive care assessment mechanism, after evidence of significant associations between the COVID-19 pandemic and these environmental indicators [10]. Other urban studies have also established links between how the pandemic has changed human behavior and how this has impacted the different environmental indicators in multiple cities around the world [11–15]. These environmental monitors are a critical component of urban and territorial management and development [16,17].

This study focuses on the development and deployment of an environmental monitoring platform framework for urban studies through low-cost sensors for smart and sustainable cities in developing countries. We propose an architectural design that connects different environmental sensors to an open data platform, proposing a framework on how the Internet of Things (IoT) can be used for pollutant monitoring for urban studies, targeting territorial, social and environmental sustainability. The study aims to initiate a discussion on how platforms can create awareness and motivate citizens to have a healthier lifestyle, taking into consideration the effect of pollution in their life and the surrounding territories. In addition, we seek to understand the behavior of the pollutants during a pandemic crisis, such as COVID-19, and their impact on a landscape. To achieve this, the platform was deployed in two cities of the Dominican Republic—Santo Domingo and Santiago de Los Caballeros—becoming the first real-time environmental monitor in the country.

This paper is organized as follows: In Section 2, a review of the literature is explored, building on the different dimensions that form our proposed framework. Section 3 presents the technology used to build the platform and its integration. In Section 4, we present our exploratory analysis on the behavior of the pollutants. We present two case studies and discuss the implications of the platforms. Finally, in Section 5, we summarize the article and present the limitations of our work. Furthermore, we highlight the most relevant findings and propose interesting questions for future research.

## 2. An Environmental Monitoring Platform for Urban Studies in a Smart and Sustainable City

The link between health complications and poor air quality or prolonged exposure to noise level has been extensively established [8,18–21]. Citizens' awareness of pollution levels and how to adapt their behavior as a response has become a new focus. In Oslo, Norway, a group of sensors were deployed in kindergartens to plan children's outdoor activities depending on the pollution levels [22]. Kumar et al. [23] measured the impact individual vehicles had during drop-off and pick-up time at a school compared to other times of the day by measuring the air quality at different points in the school with the purpose of promoting commuting in their student population. This shows the importance of environmental pollution awareness among citizens of a smart and sustainable community.

Some regions have been monitoring their air quality for multiple decades [24]. These platforms have been useful for developing prevention plans and ensuring the compliance of factories or industries with regard to their environmental regulations, to reduce the impact they have on a landscape [25,26]. In addition, monitoring air quality and noise level have been helpful in linking them with changes in human activity and their territorial expansion. Xu et al. [15] observed an improvement in air quality in different Chinese cities during confinement imposed because of the COVID-19 pandemic. Similar observations were established in multiple cities of India [14], Brazil [13] and the United States [18].

### 2.1. Internet of Things (IoT) in Environmental Monitoring for Urban Studies

To obtain the environmental data needed to observe changes in the contaminants, multiple monitoring stations are needed. Through IoT, these environmental sensors generate massive amounts of data that, if used properly, can enhance objective decision making. However, detailed observations of air quality on an urban scale are rare given the high cost of traditional monitoring stations [22], especially in developing countries.

Advances in sensor technology have allowed for the development of low-cost measurement equipment for the observation of small-scale spatial variability of pollutants [27].

These sensors, being smaller and cheaper, allow for the deployment of a higher-density network in urban spaces through IoT [28]. This advancement has allowed for the expansion of existing platforms for permanent [24] or transitory purposes on time-specific needs [29]. Besides the cost, these internet-enabled sensors allow for fast deployment and a customizable array of environmental variables depending on the requirements of each territory [30].

These IoT devices utilize electrochemical probes to detect, in real time, the concentration level of a gas. The material of the probe is designed to have a specific reaction depending on the pollutant and its concentration. For measuring particle concentration, different algorithms are used to determine the attenuation of a light signal due to the size of the particles [28]. However, the accuracy of these techniques has been questioned.

The state of development of these devices has improved significantly since their first introduction. A push from the industry and academia has improved both the accuracy and reliability of the sensors through studies that focus on understanding their performance under laboratory conditions and field studies [31]. Under laboratory conditions, these monitors tend to be more accurate than in the field [28]. If the sensors are used outside, they have better performance when exposed under higher concentrations of the contaminants [32]. With proper calibration, the IoT sensors can improve their performance and accuracy. However, these devices must be calibrated individually, since a unified calibration factor has not been identified [33]. New studies are focusing on understanding the long-term performance of these sensor networks not only to understand the impact the weather has on their performance, but also to assess how their performance changes through time [17].

Combining IoT with artificial neural networks (ANN) enables the possibility of predicting and forecasting environmental conditions at unmeasured points to create high-resolution pollution maps [34]. Using ANN for forecasting pollution reduces the complexity in understanding the interaction of each pollutant with other variables [35]. Since ANNs are problem-specific, a model cannot be easily replicated at other points or with other pollutants [34]. However, this is not a big inconvenience since sensor calibrations must be carried out individually [33].

### 2.2. Engaging with Environmental Data

Through an environmental monitoring platform, researchers have looked for the opportunity to establish a link between the landscape and the community so that they could monitor the contaminants that have a direct impact on their daily lives. This implies that engaging with different actors of society is important to obtain a holistic view and to better understand what problems are affecting the community and their surrounding territories. In turn, this will allow interventions to be established that have positive behavioral changes [36]. With this focus, different perspectives have been proposed and a citizens science approach has been widely accepted.

Using this approach and environmental monitoring through IoT can help new services to be developed for the benefit of a community by creating new partnerships [37]. This involvement helps us to understand the technology that forms the platform and develops trust [38], facilitating the adoption of the tools and the influence it has in the citizens' behavioral change. The communities that are more aware of their level of pollution exposure are more conscious of the health risks associated with that exposure [5] and their effect on the landscape.

Multiple studies have capitalized on the citizens science approach to engage with the community by promoting interactive quizzes and workshops [39], active participatory monitoring campaign [40,41], easy-to-understand maps [21], attractive visualization platforms [42], and mobile applications [43] to make sense of the environmental data.

## 2.3. Open Data

Open Data is defined as data that are available, accessible, and can be re-used and redistributed by anyone. This type of data must be available on a commonly used and machine-readable format. The license of the data must be allowed for re-use and redistribution so that the data can be repurposed for other services. Finally, there should be no restriction on who can use the data [44].

In a push for public sector transparency, governments have led the creation of Open Data Ecosystems with a government-created dataset as a shared resource. These ecosystems are valuable for government departments that can share the cost of data production and distribution. From a practical point of view, the success of open data arises when there is a collaboration between different actors and the use of the datasets [45]. There are multiple studies that describe the economic value of open data [45,46] as well as the social value that it creates [47].

From the context of smart and sustainable cities, an open data ecosystem is crucial for its development and engagement with citizens [48]. To develop the ecosystem, it is important to consider the movement of resources where demand encourages supply to create a sustainable relation between data suppliers, intermediaries and users [49]. Open Data go beyond accessibility and push towards capacity-building and training to assure the usage of the datasets. An Open Data Ecosystem incentivizes the creation of knowledge, value creation and self-development by providing data as a shared resource, pushing society to the next phase of super smart society, or Society 5.0 [45].

The purpose of deploying a network of sensors is to obtain real-time observations of the pollutants in a territory. Through an Open Data platform, citizens can engage with the environmental data to create value through new urban services.

## 2.4. Design and Implementation Framework

For an environmental monitoring platform for urban studies to successfully engage with the community, a holistic approach must be followed. Guided by the presented literature, this study proposes a framework with four dimensions: First, an internet-enabled network of sensors is needed. These sensors must be customizable so they can adapt to the requirements of the place where they are deployed and the capacity to measure the variables of interest [17,22,36]. The data collected must be placed in a reliable and secure infrastructure that is easily accessible by the community. Therefore, open data are considered as another dimension [50,51]. For the general public to make sense of the data, creating interactive, easy-to-understand visualizations of the territories impacted is important and used as a third dimension [52]. Finally, for citizens to go beyond the provided solutions, they need to develop the capacity to use the data and create innovative solutions for their own needs. Based on this perspective, the last dimension focuses on capacity building [3,16,43,53], as shown in Figure 1.

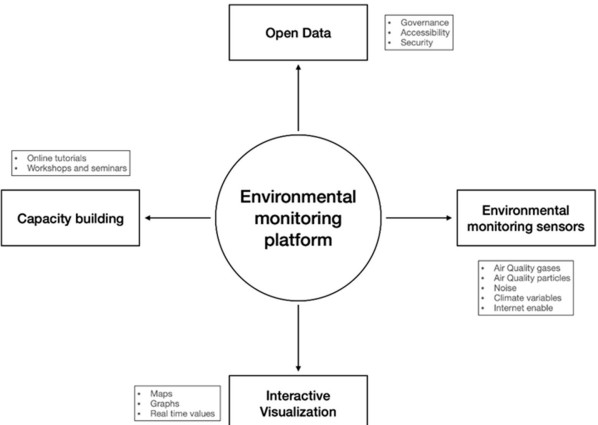

**Figure 1.** Multi-dimensional framework for an environmental monitoring platform.

## 3. Methodology

The platform was designed for the Dominican Republic, a country located in the Caribbean, forming part of the Greater Antilles. The country occupies the eastern part of the island of Hispaniola. The Dominican territory is divided into 32 provinces, of which the city of Santo Domingo is its capital and biggest city, and the city of Santiago de los Caballeros is the second largest in terms of population and economic contribution. The city of Santo Domingo has a population of more than 2 million inhabitants. The second largest is Santiago, a city in the center of the country, with great industrial and agricultural development and a population of approximately 1.5 million [54]. The platform was designed following the architecture described in this section.

### 3.1. Open Data platform

Our Open Data platform is based on a representational state transfer (REST) architectural pattern for application program interface (API) implementation. This is a popular approach in the development of web services, since it breaks one transaction into smaller parts, providing greater flexibility compared to other architectural patterns. This implementation utilizes multiple methodologies of the HTTP/HTTPS protocol, facilitating interactions with different programming languages [55]. The platform communicates with multiple services through the application programing interface (API) implemented. Some of these services come from the visualization platform, the citizens browser, our environmental monitoring sensors or other IoT devices. The development of the Open Data platform was released under a GNU GENERAL PUBLIC LICENSE and the code is available on GitHub [56].

For a GET request, the service accepts two parameters to delimit time and one parameter to specify a format. If no parameters are sent to the API, by default, the response will include all the data available in the JavaScript Object Notation (JSON) format. The platform can respond with a .CSV file if specified in the format parameter. If a specific timeframe is wanted, the platform supports start- and end-time parameters.

When making a POST request, the service must authenticate with the platform when sending the JSON with the information to be saved in the database. The device which authenticates the server responds with a "201 Created" if the data are saved correctly, or "204 No Content" if an error occurs. In case there is a failed authentication, the server responds with "400 Bad Request".

### 3.2. Environmental Monitoring Sensors

The environmental IoT sensor was divided into two layers: the hardware layer and the software layer. The hardware layer is based on the Libelium's Plug & Sense! line of products [57]. These are commercially available, tested, and calibrated environmental monitoring devices. The devices encapsulate a programmable Arduino-based microcontroller board in a waterproof and weatherproof plastic enclosure. All the sensor probes plug into the enclosure via a nine-pin waterproof connector, which facilitates initial deployments and future replacements. Figure 2 depicts the physical and virtual nodes in the system, along with their links and corresponding internal software components.

The network of sensors was deployed in a university campus in Santo Domingo and in Santiago de Los Caballeros. In both campuses, there was access to a reliable Wi-Fi network. To reduce the operational cost of the network of sensor infrastructure, we configured the devices to connect through the Wi-Fi___33 network of the university. For the monitoring sensor network, an exclusive service-set identifier (SSID) was configured. Having an exclusive SSID increases the security of the devices, as the network is only shared with the devices that connect to the platform.

The environmental monitoring platform, through the air quality IoT devices, measures four gaseous variables: carbon monoxide (CO), sulfur dioxide ($SO_2$), ozone ($O_3$), nitrogen dioxide ($NO_2$); three particle-matter variables: ultrafine particulate matter (PM1), fine particulate matter (PM2.5), and coarse particulate matter (PM10). These measurements

are complemented with weather measurements: temperature, relative humidity, and barometric pressure. In the case of the air-quality-monitoring devices, the measurements of each variable were obtained every twenty minutes.

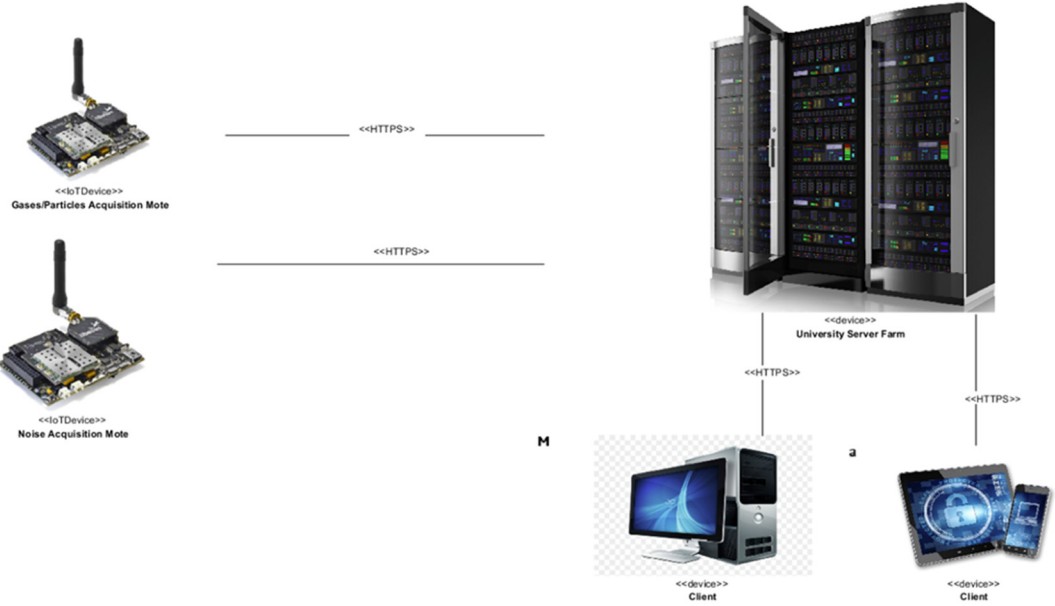

**Figure 2.** Deployment diagram of the physical and virtual nodes that compose the system.

To measure noise, the platform uses the noise IoT devices, which are equipped with high-caliber microphones. This variable is monitored every ten minutes following the IEC 61,672 standard, which obtains several noise data samples and averages these over the time-window in which the noise was produced. These devices also measure the complementary weather variables. All IoT devices track internal performance data. Table 1 shows the relation between variables, measuring unit and measuring period.

**Table 1.** Environmental variables with the unit and period in which they are measured.

| Variable | Unit of Measurement | Measurement Period |
|---|---|---|
| Carbon monoxide (CO)<br>Sulfur dioxide ($SO_2$)<br>Ozone ($O_3$)<br>Nitrogen dioxide ($NO_2$) | Parts per million (ppm) | 20 min |
| Ultrafine particulate matter (PM1)<br>Fine particulate matter (PM2.5)<br>Coarse particulate matter (PM10) | Micrograms per cubic meter ($\mu g/m^3$) | 20 min |
| Temperature<br>Relative humidity<br>Barometric pressure | Degrees Celsius (°C)<br>Percentage (%)<br>Pascal (Pa) | 20 min |
| Noise | Additive decibels (dBA) | 10 min |
| Battery capacity<br>Battery voltage<br>Battery charge current | Percentage (%)<br>Volts (V)<br>Milliampere (mA) | 20 min |

The software layer of the IoT devices was customized for the requirements of our platform. Each device measures their assigned variables following the period established in Table 1. After the data are acquired, the device sends the data to the Open Data platform via HTTP POST following the procedure mentioned in Section 3.1. After this event, the

software agent goes to sleep and wakes up after the specified time period has elapsed. Figure 3 depicts the interaction between the software agents and the Open Data platform.

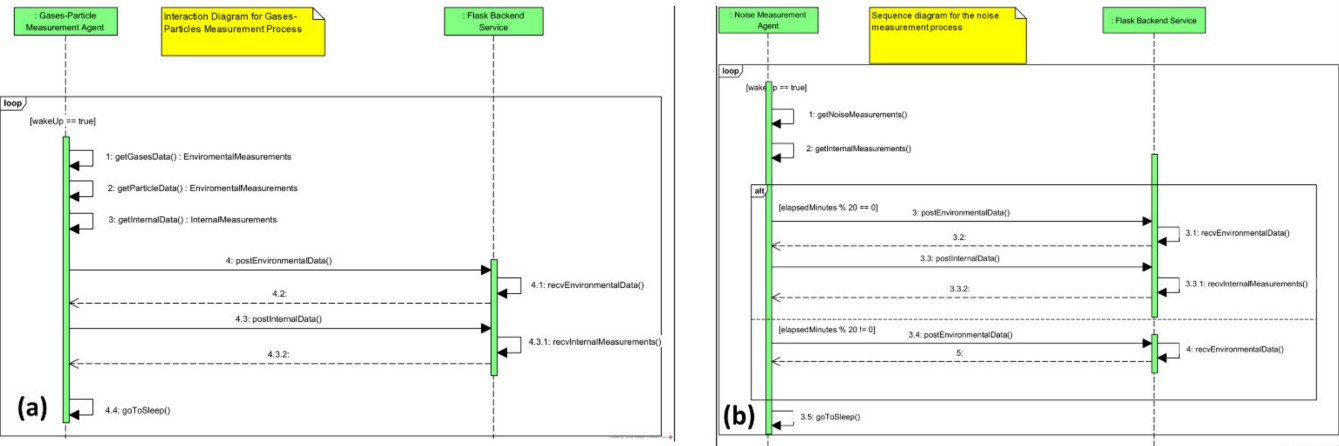

**Figure 3.** Sequence diagram between the OD platform and the IoT devices: (**a**) air quality IoT device; (**b**) noise IoT devices.

### 3.3. Determining Air Quality Index

The Air Quality Index can be determined for a short- or long-term duration [58]. The most popular algorithms follow a short-term duration, the Air Quality Index (AQI) algorithm developed by the United States Environmental Protection Agency [59] and the Common Air Quality Index (CAQI) developed in order to make the levels comparable between the cities of the European Union [60,61]. Both algorithms are similar in how they determine the index—they perform a linear interpolation between the edges of the classes. In the case of AQI, the classes are determined by the regulations of the United States, and in CAQI, by the regulations of the European Union. For each pollutant, a sub-index is determined and the final index is the highest sub-index. To determine each subscript, the following Equation (1) is used:

$$I = ((Ih - Il)/(Ch - Cl)) (C - Cl) + Il, \tag{1}$$

where

I　is the index to determine;
C　is the reading of the concentration of the contaminant;
Cl　is the bottom edge of the class where C is found;
Ch　is the top edge of the class where C is located;
Il　is the value of the index that corresponds to Cl;
Ih　is the value of the index that corresponds to Ch.

These indices have a short-term advantage when there is an increase in any of the pollutants, allowing the population to be alerted quickly [62]. The index does not need to consider all pollutants, so a platform can have sensors with different variables deployed within a city. The environmental monitoring platform implemented uses the AQI algorithm, since the Dominican Republic has similar regulations to the United States. The variables considered to determine the index are carbon monoxide (CO), nitrogen dioxide ($NO_2$), ozone ($O_3$), sulfur dioxide ($SO_2$) and particulate matter in two sizes (PM2.5 and PM10).

### 3.4. Interactive Visualization

This dimension should allow for the easy interaction of environmental data for exploration and analysis to a layperson user. For this purpose, three data dashboards were developed to simplify the interaction with multiple variables.

The Air Quality Dashboard was divided into three parts. The first part presents the last measured variables, using a gauge to represent the current air pollution level following the Air Quality Index (AQI) algorithm. This graph is followed by the current values of particulate matter at 1, 2.5 and 10 μm and the concentration of carbon monoxide, nitrogen dioxide, ozone and sulfur dioxide at ground level, as shown in Figure 4.

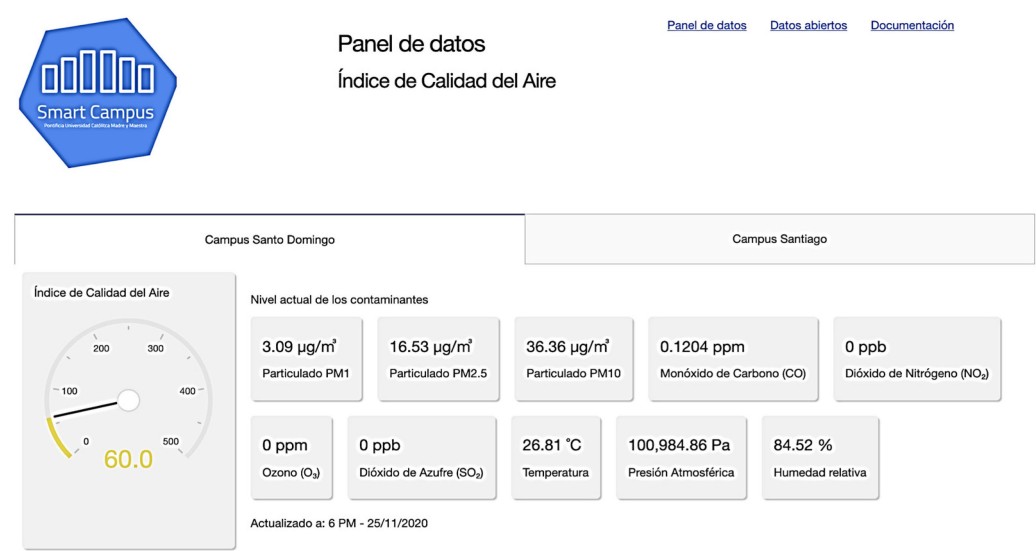

**Figure 4.** View of the Air Quality Dashboard, the last measurement section.

The second part of the dashboard is the exploration control panel. In this panel, the user can indicate the locations and the date range for the analysis. In addition, the user can manipulate how to aggregate the data for the analysis. Following the AQI algorithm, the data can be aggregated per month, week, day or hour.

The third part of the dashboard graphs the Air Quality Index along with the different contaminants according to the parameters selected by the user. All the graphs automatically update as a user changes any of the parameters. As an example, Figure 5a shows a time series of daily aggregate AQI from one point for December 2020, and Figure 5b shows a time series of hourly aggregate AQI from one point for the first seven days of January 2021.

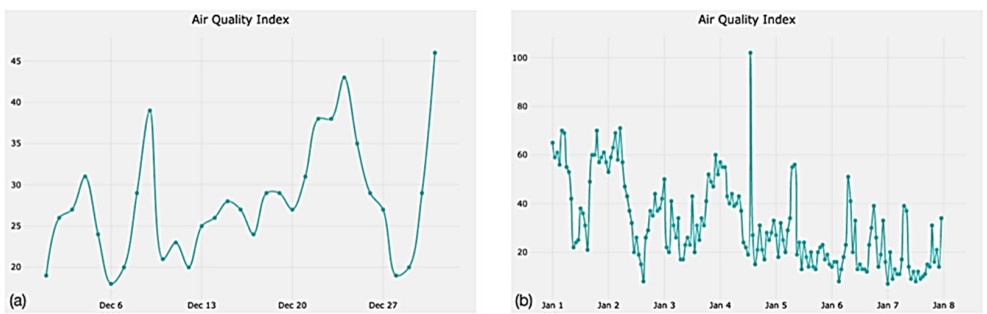

**Figure 5.** (**a**) Daily aggregate AQI time series for the Santo Domingo Campus Monitor for December 2020; (**b**) hourly aggregated AQI time series for the Santo Domingo Campus Monitor from 1 to 7 January 2021.

The second dashboard focuses on noise level and follows a similar design pattern to the Air Quality dashboard. As an example, Figure 6 shows an hourly aggregated noise level comparison of three monitors for the first work week of January 2021. The third dashboard shows the weather variables; the user can change the parameters following the same procedures of the two previous dashboards.

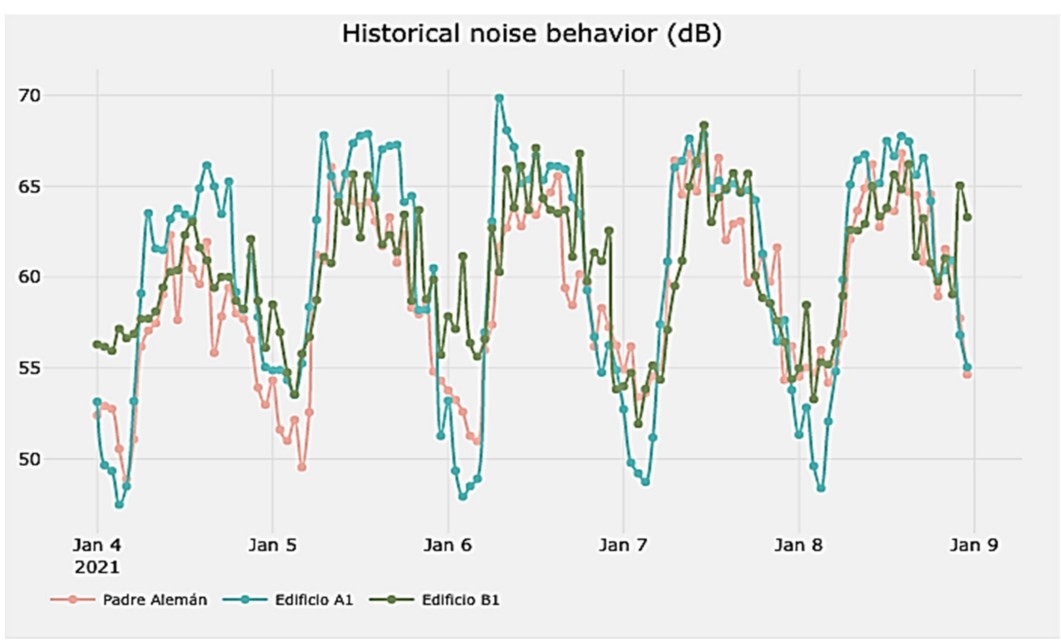

**Figure 6.** Hourly aggregated noise level time series between 4 and 8 January 2021.

### 3.5. Capacity Building

To motivate the usage of the platform, multiple strategies were developed. First, different tutorials were published on the website of the platform in the form of short videos and written explanations. These focused on how to use the different dashboards, the platform's API and the technology behind the IoT devices used to capture pollution levels [63]. In addition, we explain how this initiative aligns with the United Nations Sustainable Development Goals as a course of action for attracting new partners to expand the network of sensors. Furthermore, different workshops and seminars were developed targeting a wide range of audiences by customizing the content to their interests.

## 4. Results and Discussion

This section presents two case studies showing how the platform can be used to create awareness in a community with regard to the effect of pollution. For the first case study, we used data obtained from the platform limiting the dates between 22 and 24 June 2020. For the second case study, we extended the date range from 18 May 2020 to 26 January 2021.

### 4.1. Case 1: Sahara Dust

In this case study, we analyzed the impact of dust landscapes from the Sahara had on the air quality in Santo Domingo in 2020. This phenomenon is a recurrent event that happens every year, where the atmospheric difference in the Sahara Desert loads the dry air with fine particles that travel thousands of kilometers by air towards America. This phenomenon is known to be a health risk to the population as it can cause multiple respiratory issues, especially for sensitive groups [64].

The citizens of Santo Domingo experienced this event in 2020 between 22 and 24 June. On 22 June, the daily average Air Quality Index was 159 and, on 24 June, it was 153—both are considered to be unhealthy scores. The peak was reached on 23 June, where the daily average was 205, considered very unhealthy. This platform was able to register, for the first time, changes in the air quality of the Dominican Republic. Figure 7 shows the air quality fluctuation per hour while the Sahara dust cloud crossed the city. Figure 8 displays the fluctuation per hour of particles and gasses associated with this territorial dust. Both graphs provide a clear indication of the impact the Sahara dust phenomenon has on the air quality index, and how these fluctuations are dangerous for those who are exposed.

Even for short exposures, this level of contaminant can have a negative impact on the population's health and on the landscape.

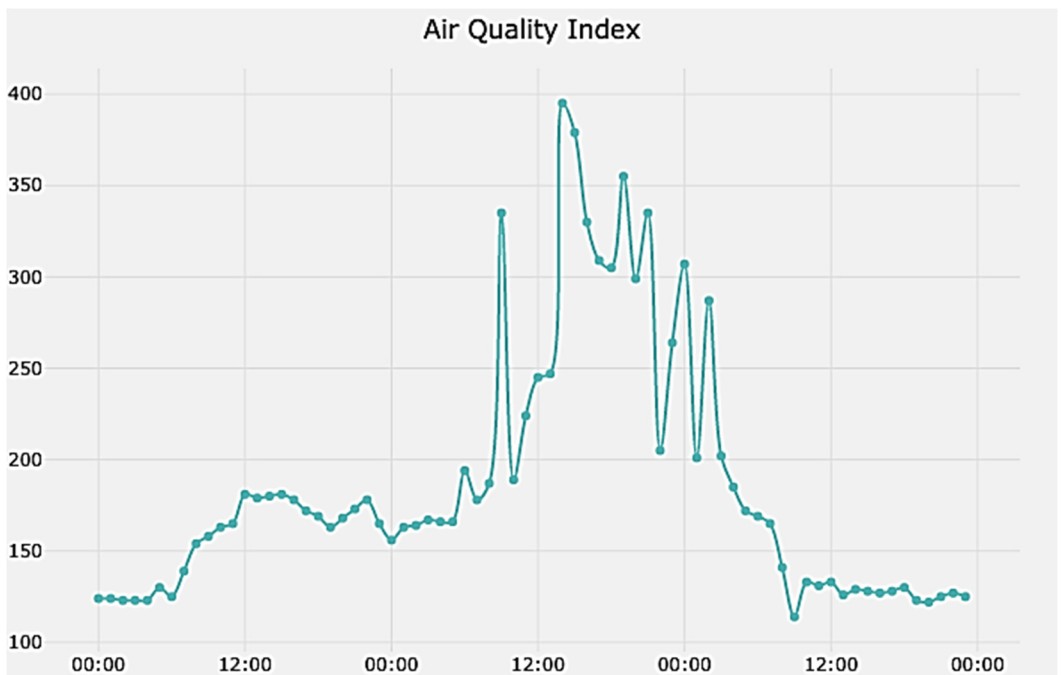

**Figure 7.** Hourly aggregated air quality index time series between 22 and 24 June 2020.

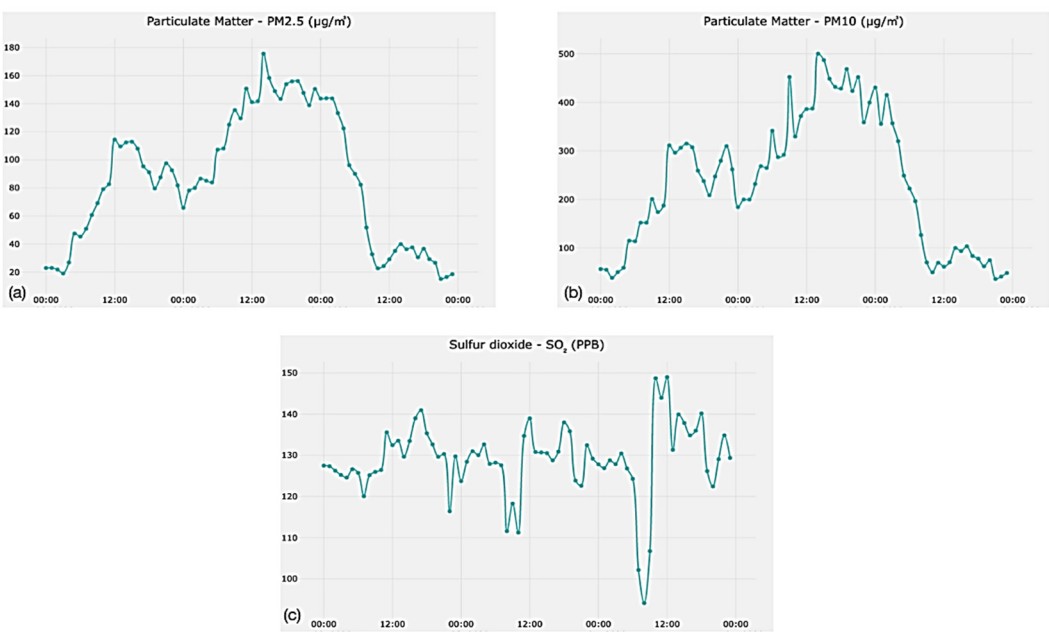

**Figure 8.** Hourly aggregated (**a**) fine particulate matter, (**b**) coarse particulate matter and (**c**) sulfur dioxide time series between 22 and 24 June 2020.

After the event, the platform was used to create awareness of the danger of the particulate matter to the respiratory system; multiple newspapers referenced the platform as a tool to monitor environmental conditions. The platform marks the first time that the changes caused by this event were registered. In future events, a comparison of magnitude will be possible. This allows for the community to gain an understanding of the changes in the territory and plan multiple urban studies.

*4.2. Case 2: COVID-19 in Santo Domingo and Santiago de Los Caballeros*

This case study aims to understand how the policies taken to prevent the spread of COVID-19 affect the air quality and noise level of the landscape and community of the two most important cities in the Dominican Republic. The first positive case of COVID-19 reported in the country was on 1 March 2020. With the first case, the Dominican government focused on preventing the spread of the virus in the country. Among the measures taken, the government declared a state of emergency and issued a nationwide curfew, affecting the two cities taken into consideration in this case study. From the first decree, establishing the curfew, the time range of the curfew was either reaffirmed or modified via decrees. Table 2 shows a summary of the changes in the time of the curfew by decree in the time frame of this study. All decrees found on the table were obtained from the Office of Legal Consult of the executive branch of the Dominican Republic [65].

**Table 2.** Curfew schedule established by the Dominican government.

| Period | Decree | Start Date | End Date | Weekday Schedule | Weekend Schedule | Free Mobility |
|--------|--------|------------|----------|------------------|------------------|---------------|
| 14 | 007-21 | 11 January 2021 | 26 January 2021 | 5:00 p.m.–5:00 a.m. (+1) | 12:00 p.m.–5:00 a.m. (+1) | 3 h |
| 13 | 740-20 | 1 January 2021 | 10 January 2021 | 5:00 p.m.–5:00 a.m. (+1) | 12:00 p.m.–5:00 a.m. (+1) | 2 h (weekdays) |
| 12 | 698-20 | 15 December 2020 | 31 December 2020 | 7:00 p.m.–5:00 a.m. (+1) | 7:00 p.m.–5:00 a.m. (+1) | 3 h |
| 11 | 684-20 | 2 December 2020 | 14 December 2020 | 9:00 p.m.–5:00 a.m. (+1) | 7:00 p.m.–5:00 a.m. (+1) | |
| 10 | 619-20 | 12 November 2020 | 1 December 2020 | | | |
| 9 | 554-20 | 18 October 2020 | 11 November 2020 | | | |
| 8 | 504-20 | 28 September 2020 | 17 October 2020 | | | |
| 7 | 431-20 | 3 September 2020 | 27 September 2020 | 7:00 p.m.–5:00 a.m. (+1) | 5:00 p.m.–5:00 a.m. (+1) | |
| 6 | 298-20 | 9 August 2020 | 2 September 2020 | | | |
| 5 | 266-20 | 21 July 2020 | 9 August 2020 | | | |
| 4 | No curfew | 28 June 2020 | 20 July 2020 | | | |
| 3 | 214-20 | 14 June 2020 | 27 June 2020 | 8:00 p.m.–5:00 a.m. (+1) | 8:00 p.m.–5:00 a.m. (+1) | |
| 2 | 188-20 | 2 June 2020 | 13 June 2020 | 7:00 p.m.–5:00 a.m. (+1) | 7:00 p.m.–5:00 a.m. (+1) Sat 5:00 p.m.–5:00 a.m. (+1) Sun | |
| 1 | 161-20 | 18 May 2020 | 1 June 2020 | | | |

4.2.1. The Effect of the Curfew on Noise

The curfew has a clear effect on societal activities and the impact these activities have in surrounding territories. Several of these activities have shifted in time—businesses such as bars and restaurants were not able to have customers on site due to the state of emergency.

Using the Noise IoT devices, a variation in daily noise profile was detected. The data suggest the noise could be used to determine the levels of activity in an area. Most of the noise level is related to traffic, but a special situation is given in one facility located 60 m from a small commercial area consisting of several bars and restaurants. Figure 9a,b compare the average noise level by hour in two different periods at this location. During period 4, the state of emergency and any restriction was suspended. In period 5, the state of emergency and the curfew were re-established—the new restrictions allowed restaurants and bars to operate with customers on site; however, their operating hours were made to adapt to curfew schedules.

The graph shows an important difference with respect to night-time noise level, suggesting the activity of restaurants and bars. Additionally, we observed a peak during the curfew in the early afternoon during weekdays and around noon on weekends. This peak could be related to the concentration of citizens' activities as the curfew start time approached.

When the curfew time changed, a similar behavior was observed. Figure 9c,d show a comparison between periods 7 and 8. Period 8 has a more relaxed curfew by adding two hours for commercial and social activities. At the same location, the noise data clearly reflect how the new curfew prolonged the operating hours of the nearby businesses. This

shows that using the platform behavioral changes of citizens can be monitored and analyzed, creating a valuable tool for a community when creating policies or studying the implication of a policy.

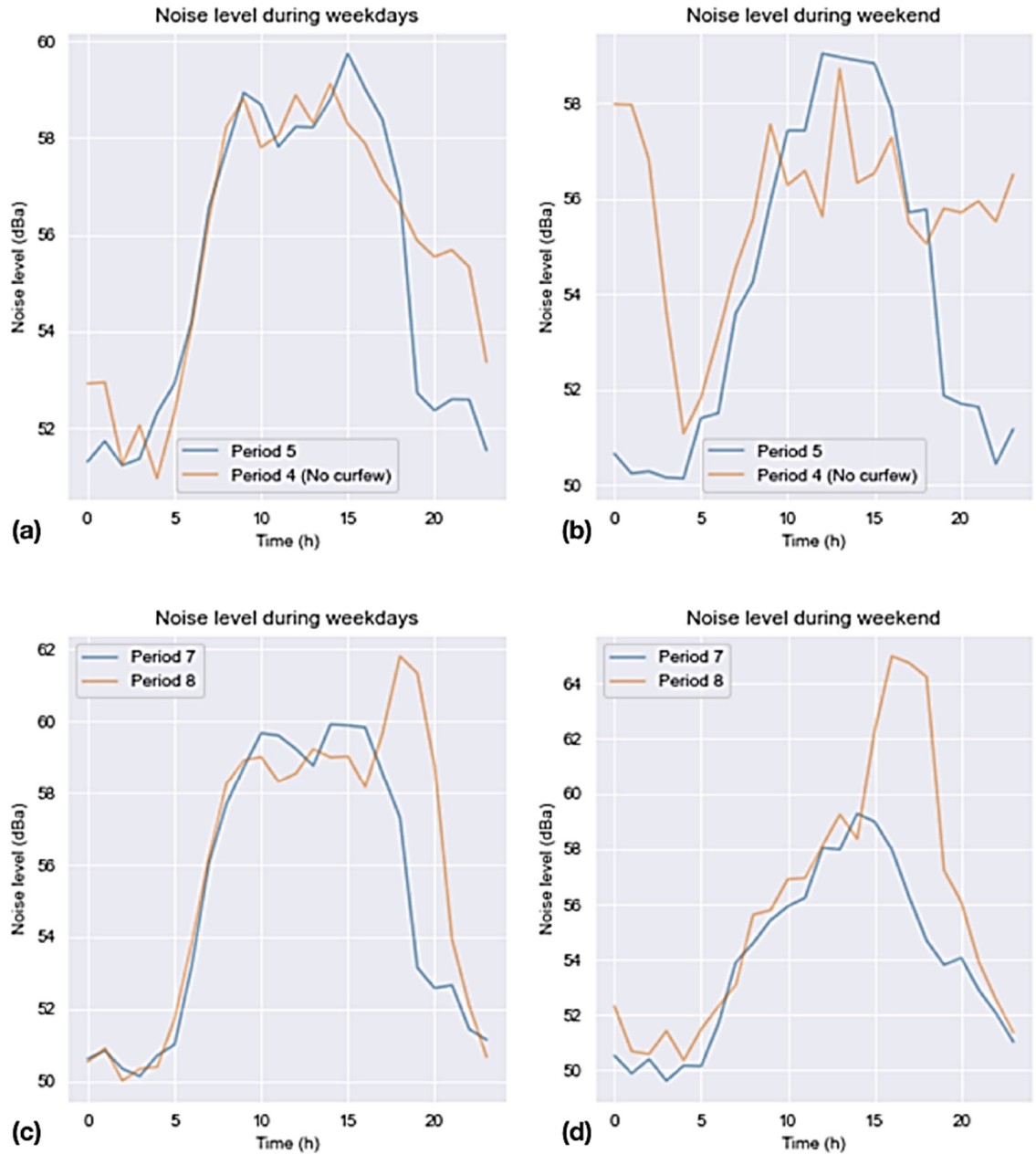

**Figure 9.** Comparison of the average noise level by hours (**a**) during the weekdays between a period with curfew and one without, (**b**) during the weekend between a period with curfew and one without, (**c**) during the weekdays between periods with different curfew schedules and (**d**) during the weekend between periods with different curfew schedules on a commercial area.

4.2.2. The Effect of the Curfew on Air Quality

In Santo Domingo, the level of air quality improved as the curfew was extended throughout the COVID-19 pandemic, as shown in Figure 10a. Comparing the different pollutants, we can see that the high levels of air quality index were due to dust particles (Figure 10b) in the first three months of the study and, in the latest month, were driven by carbon monoxide and nitrogen dioxide concentrations, as shown in Figure 10c,d.

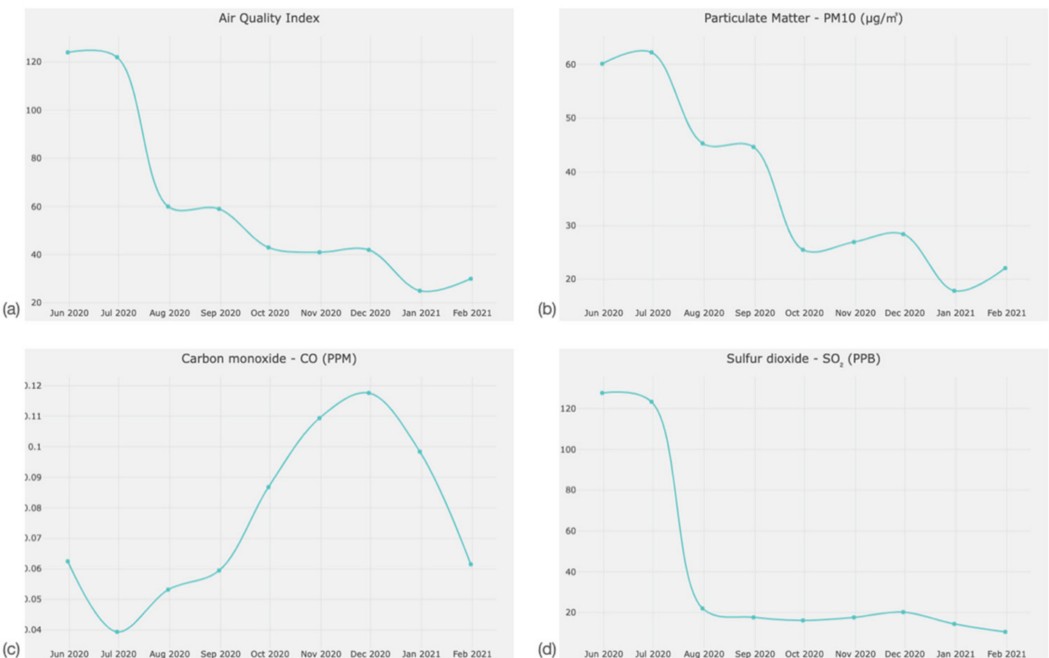

**Figure 10.** Monthly average in Santo Domingo of (**a**) air quality index, (**b**) PM 10, (**c**) carbon monoxide and (**d**) nitrogen dioxide as presented from the platform.

With the exception of naturally occurring events, a relationship can be seen between the curfew and pollutants. Figure 11a shows the hourly behavior of the air quality index from 28 to 29 September 2020. These days were chosen as they represent the citizens' behavior when the curfew was relaxed in period 8. The air quality index increased when the curfew ended and the index decreased as the curfew started. Comparing carbon monoxide levels during the strictest curfew, the weekends of period 13, Figure 11b shows how the concentrations of this gas increased when the curfew finished and decreased when it started between 2 and 3 January 2021.

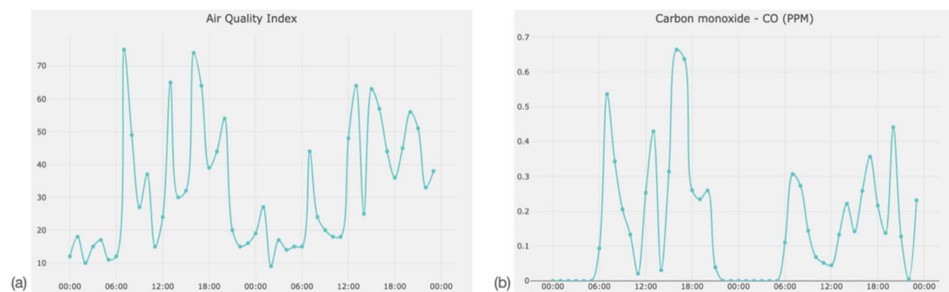

**Figure 11.** Hourly average in Santo Domingo of (**a**) air quality on 28 and 29 September 2020 and (**b**) carbon monoxide on 2 and 3 January 2021.

Similar to in the city of Santiago de Los Caballeros, the data suggest that a more relaxed curfew increased the level of pollution, reducing the air quality. Figure 12a shows the variability of certain gasses and fine particulates across the last seven periods. Sulfur dioxide average levels fell relatively slightly when compared to the rate at which carbon monoxide rose during the same periods. Similar to the levels of sulfur dioxide, the particulate matter data exhibit a downward tendency during the same period. Figure 12b shows how contamination associated with both types of particulate data becomes minimal specifically during the ninth and tenth periods and increases during the last two periods.

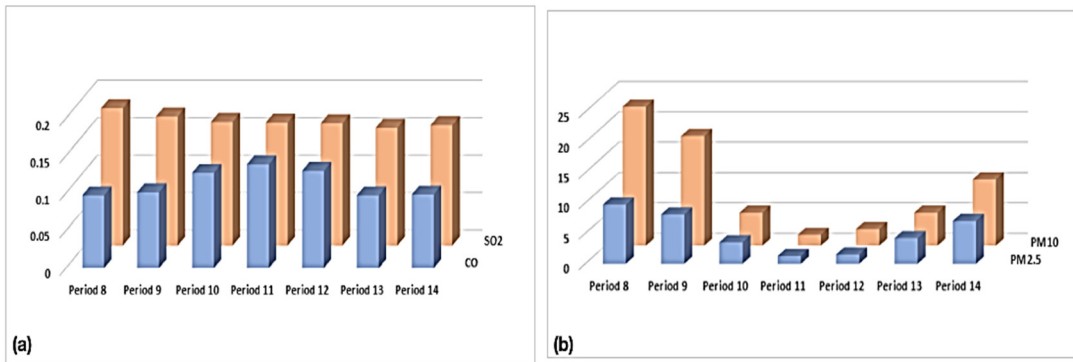

**Figure 12.** (**a**) Gas and (**b**) particulate pollution levels comparison in Santiago de Los Caballeros from period 8 to 14.

A further analysis of the behavior of carbon monoxide, a gas related to motor vehicle activities, during each day of the week shows an increase between periods 10 and 12. However, on average, the most contaminated weekday does change throughout all periods. In Figure 13a, the carbon monoxide averages reach the highest value during all measured Wednesdays. However, the highest variation is observed on Thursdays. The latter may signal that the average contamination on this weekday could have occurred in short bursts in which carbon monoxide levels rapidly rose and fell. This behavior was also observed, albeit in a less dramatic manner, on the measures taken over on Mondays.

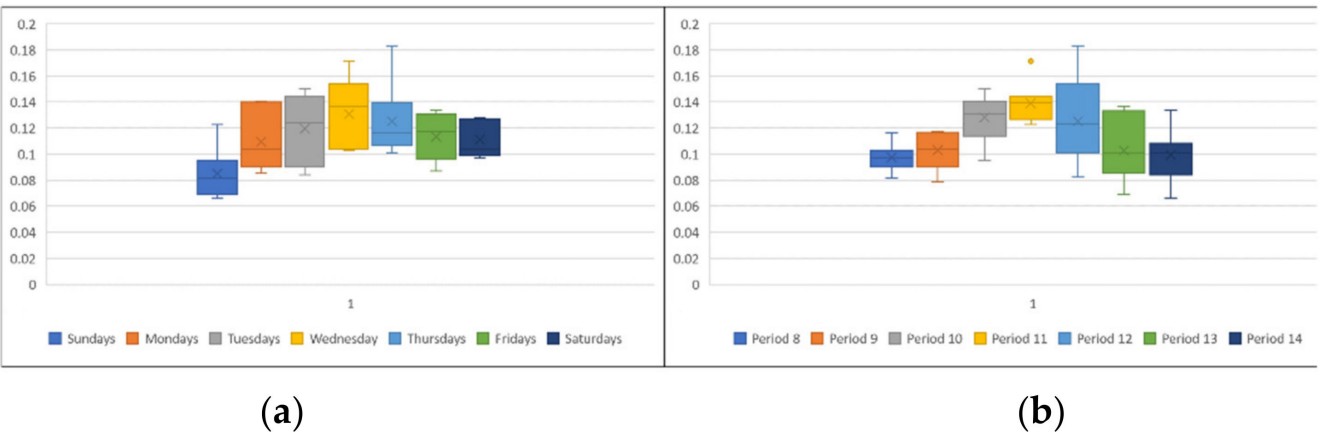

**Figure 13.** Carbon monoxide minimum, maximum and average over (**a**) all weekdays and (**b**) periods in Santiago de Los Caballeros from period 8 to 14.

As shown in Figure 13b, the 11th period exhibits the highest average among all other periods. We can also see that the maximum in this period (0.1714 ppm) is a possible outlier as it falls outside the quartile in which the maximum should have fallen. This could be due to exceedingly high levels of pollution during relatively short periods.

Similar to noise, the air quality aspect of the platform can be used to assess the pollutant levels and their changes as policies are implemented and how they impact a landscape. With real-time monitoring of the pollutants, a community can better understand the impact human activity has on the environment and the territories they occupy.

## 5. Conclusions

There are many challenges when developing and deploying an environmental monitoring platform for air quality and noise level. On one hand, the platform must be flexible to interact with different environmental variables, accessible so that a community can easily use it, and secure to establish trust. Another aspect is how to interact directly with the community to create awareness of pollution and the impact they have in surrounding

territories. Different strategies are required to reach the community. In this work, we present a framework for developing such a platform and how it can be used to understand both natural phenomena and human behavior impact on the pollutant and our territories.

Two case studies were presented; the first case study focused on how the platform can be used to understand the impact a natural event, for example, how dust landscapes impact a community and the actions that can be taken. This study analyzes the effect of the Sahara dust in Santo Domingo in 2020. The results clearly portray an increase in the Air Quality Index as the phenomenon crosses the city. An awareness campaign after the event allowed for citizens to understand the phenomenon and the danger of exposing themselves to the dust, incentivizing them to take action. With a monitoring platform, such as the one presented in this article, the effect of the phenomenon is recorded and allows for a comparison with future events for both policymaking and citizens' awareness. Multiple urban studies could be developed following these data.

The second case study portrays how changes in policies can have a direct effect on the pollutants that affect the community and landscape of a city. This enquiry explored how the policies taken to prevent the spread of COVID-19 in the Dominican Republic affected the air quality and noise level of Santo Domingo and Santiago de Los Caballeros, the two biggest cities in the country. The analysis showed an improvement in the air quality and a decrease in noise level as the curfew established by the government was stricter, limiting the freedom of movement in the population. As the restrictions were relaxed, an increase in the level of pollutants was recorded. This study reflects on the importance of environmental monitoring platforms for understanding the impact different policies have on the pollutants that surround a community and the landscape.

In the long term, environmental monitoring platforms allow for the assessment and impact of different recurring phenomena through time. The open access of the data encourages a culture of data-driven decision making, which improves the decision processes of a community. With the second case study, we were able to show how a curfew policy used for controlling the spread of COVID-19 affected the levels of air quality and noise in an urban space. If policymakers are data-driven, incorporating this dataset can help establish a holistic view that shows the effect the policy has on other aspects of community and the landscape.

An awareness campaign for the platform was launched where multiple actors showed interest in learning about the pollutant concentration, how to access the data and how to use the visualization apps. Even if the visualizations are thought to be intuitive, short sessions or tutorials are needed to reach a broader audience. A common concern was focused on the geographical area covered by the network of sensors and the expansion of the network in other localities. When deploying this type of platform, it is important to plan for geographical coverage and future expansion. The cost of sensors and their deployment is a major factor to consider in the expansion of the platform. The introduction of low-cost sensors allows for a tighter grid of measurement and a better understanding of how smaller communities in an urban space are affected by the pollutants.

*Limitations and Future Work*

This research is consistent with the development of smart and sustainable cities through an environmental monitoring platform for urban studies focused on air quality and noise level for cities in developing countries. However, this study has some limitations in the framework, the deployment of the platform and the case studies presented.

The framework provides different dimensions so that different components of the platform can operate. The framework does not expand on the mechanisms under which third-party applications can interact with the platform; this is an important limitation. The Open Data dimension must define the governance of the data in the platform as sensors from other partners are integrated to the platform and with the addition of new services from third parties.

Their deployment was focused on the two biggest cities of the Dominican Republic. The main limitation of the deployment is in regard to the number of sensors initially deployed, which did not cover the extension of each city. More sensors are needed to create a robust network that can measure the contaminants in each neighborhood of the cities. Furthermore, the platform is limited in terms of the functionalities related to IoT devices. To improve this issue, the platform could incorporate more lightweight application-level protocols such as constrained application protocol (CoAP). Adding this protocol to the platform would facilitate communication between the platform and low-cost sensors that are resource-constrained, which could facilitate the expansion of the platform.

Through an expansion of the platform in both sensors and functionalities, it would be of interest to investigate the value it creates on other territories and measure the trust of the community to the monitoring platform, especially those at risk. In addition, it is of interest to compare the behavior of the pollution in different parts of a city or country in order to develop more resilient communities.

From a case study perspective, multiple limitations exist. The short research period is one of those limitations. Further work is needed to understand the long-term performance of the platform, of pollutants and the usage given for other urban studies. For example, a multi-year study of the impact of the Sahara Dust in the air quality of Santo Domingo.

This study has focused on understanding the relation between urban conditions and a community and the impact they have on each other through the variables measured by the platform. This itself is a limitation, as the impact the platform has on the landscape is not considered; it is of interest in future studies to evaluate this through a life-cycle assessment of the platform.

In addition, the air quality sensors are limited to three types of particulate matter (PM1, PM2.5 and PM 10) and four gases (CO, $NO_2$, $O_3$ and $SO_2$). Even if these pollutants are the most critical for determining air quality, there are other gases that are not measured, and their impacts are not assessed on this study. Other studies should consider incorporating a broad variety of sensors to measure a broad variety of pollutants.

**Author Contributions:** Conceptualization, V.G. and J.F.-G.; methodology, V.G. and M.P.; software, V.G. and M.P.; validation, V.G., J.F.-G., M.P. and Y.G.F.; formal analysis, V.G., J.F.-G. and M.P.; investigation, V.G., M.P., J.F.-G. and Y.G.F.; resources, V.G., and M.P.; data curation, V.G.; writing—original draft preparation, V.G.; writing—review and editing, M.P. and Y.G.F.; visualization, V.G., J.F.-G., Y.G.F. and M.P.; supervision, V.G.; project administration, V.G.; funding acquisition, V.G. All authors have read and agreed to the published version of the manuscript.

**Funding:** This research received no external funding.

**Data Availability Statement:** The data that support the findings of this study are openly available in Zenodo at https://doi.org/10.5281/zenodo.7105153.

**Acknowledgments:** We would like to thank the support personnel that aided regarding the sensor-nodes' installation on site in both, Santo Domingo and Santiago de Los Caballeros.

**Conflicts of Interest:** The authors declare no conflict of interest.

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
