# Peer review of "Real-Time Environmental Monitoring Platform for Wellness and Preventive Care in a Smart and Sustainable City with an Urban Landscape Perspective: The Case of Developing Countries"

_land, doi:10.3390/land11101635_

Round 1

Reviewer 1 Report

The paper describes a system for monitoring the quality of an urban environment. The system monitors pollutants in the air (particulate and harmful gases), noise, and meteorological conditions. The authors show a prototype that implements the sensor and describe the infrastructure collecting the data.

One point of interest in the project is the open data approach. In addition to the sensor network, the authors propose and implement a web service that gives simplified access to the collected data, to foster informed planning.

The paper is well organized but its originality is limited. To improve the contribution, the authors should take into account (either suggesting a solution or just addressing for future research) a number of issues:

-) security concerns: the authentication mechanism used in the protocol (section 3.1) is not explained, but is fundamental for evaluating infrastructure security. In addition, I do not agree with the reason why HTTP(S) is popular in IoT. And, in the end, HTTP(S) is not so popular in IoT (consider CoAP);

-) source code is not provided: the article does not contain original ideas, but provides evidence of a valuable implementation. It is therefore important that the authors give other researchers access to the matured experience, by publishing on appropriate platforms the code and layouts they produced;

-) the paper does not explain the link layer technology used for the network infrastructure: WiFi, Broadband, other? This is relevant to assessing reliability and cost (see below)

-) given the cut of the article, a holistic attitude regarding side aspects is due. Does the system have an impact on the landscape? How device's cost limits its application? Is there an impact on privacy? Do the authors have an idea of the carbon footprint? Even if you do not have a complete answer to such questions, mentioning the problems in your paper helps the reader to understand that those are relevant aspects

-) the proposed solution is a partial implementation of Citizen Science (CS) since information flows one way, from the System to the Citizens. In CS the reverse direction is relevant (indeed, it is the novelty of CS). Can the Citizens participate in data management, for instance installing the sensors on their premises, or providing feedback about perceived environment quality?

Reviewer 2 Report

The paper definitely covers a very timely topic and it is of potential interest for a wide audience / readership. However, there is a list of improvements to be made to ensure it is ready for being published and deployed widely.

Abstract states "this research shows a framework", while I would strongly recommend to provide some background on the purpose of the development of this framework and, more importantly, making more obvious who developed it (probably authors), which also supposes avoidance the term "shows" replacing it by something more scientifically soundy and study-compliant, e.g., "elaborates on the framework we developed ...." OR "provides a detailed insight on a successfully developed and deployed framework...". At the same time "a framework in which 11an environmental monitoring platform was deployed" is not very accurate, please, consider rewording this sentence, also I would suggest replace IoT mentioned there with a more specific example the authors refer later - " low-cost sensors".

"Smart and Sustainable Communities" also supposes at least a brief elaboration on Society 5.0, also known as super smart society or society of imagination. There are also studies linking open data platforms, which constitute a part of the proposed framework, to smart communities and also sensor-based data available on these platforms. 

Fukuyama, M. (2018). Society 5.0: Aiming for a new human-centered society. Japan Spotlight, 27(5), 47-50.

Sołtysik-Piorunkiewicz, A., & Zdonek, I. (2021). How society 5.0 and industry 4.0 ideas shape the open data performance expectancy. Sustainability, 13(2), 917.

Nikiforova, A. (2021). Smarter open government data for society 5.0: are your open data smart enough?. Sensors, 21(15), 5204.

The latter on Smarter open data for society 5.0 reflects on the availability of these data and "readiness" for the reuse etc. This can also bring a light on the current readiness of countries (although more governmental level) to proceed with these solutions.

References to be of interest for the Section 2 are:

Roman, D., Reeves, N., Gonzalez, E., Celino, I., Abd El Kader, S., Turk, P., ... & Simperl, E. (2021). An analysis of pollution Citizen Science projects from the perspective of Data Science and Open Science. Data Technologies and Applications.

Chen, L. J., Ho, Y. H., Lee, H. C., Wu, H. C., Liu, H. M., Hsieh, H. H., ... & Lung, S. C. C. (2017). An open framework for participatory PM2. 5 monitoring in smart cities. Ieee Access, 5, 14441-14454.

Stieb, D. M., Evans, G. J., To, T. M., Brook, J. R., & Burnett, R. T. (2020). An ecological analysis of long-term exposure to PM2. 5 and incidence of COVID-19 in Canadian health regions. Environmental research, 191, 110052.

There are also interesting examples to be added to Section 2 from Spanish OGD portal, including pollution-, air quality- related ones. One of them is App Plantes -“Flora urbana y alergia, ¿cooperas?”(Urban flora and allergy, do you cooperate?) 

Considering the nature of the study, it is recommended to adapt the title making it more study-compliant, including emphasis that this is rather a use-case (country level or developing countries in general), which can be added as a complementary part of the title. Also, a discussion on generalization would be beneficial.

"Generally, the concept of open data is associated with data created by the govern- 150

ment and shared to increase transparency [16] but this is not always the case." - if it is more than only about the data, I would highly recommend to ensure you are accurate in the use of terminology, i.e. open data and open government data, since otherwise the sentence is misinterpret like you would like to say that not only government made data publicly accessible, i.e., the focus is shifted. Secondly, I would highly recommend to elaborate on the points you mention in Section 2.3. in more detail being indeed of high importance. Here, I would suggest to refer to the concept of "open data ecosystem" and especially in the context of smart cities - this will make this section more mature.

"The data obtained must be secure and accessible by the community" what exactly do you mean by security? it is not very obvious for the reader.

Considering the dimensions the authors identified, being in line with the literature, I would suggest to provide these dimensions with references. Here, for instance, one of the following studies being very in line with your dimensions and sub-dimensions, could be of value, where the second will also provide you with the external references per aspect:

Máchová, R. et al. (2017). Evaluating the quality of open data portals on the national level. Journal of theoretical and applied electronic commerce research, 12(1), 21-41.

Lnenicka et al. (2021). Transparency-by-design: What is the role of open data portals?. Telematics and Informatics, 61, 101605.

Lněnička et al. (2021). Enhancing transparency through open government data: The case of data portals and their features and capabilities. Online Information Review.

Of course, other studies can be used as a reference, but it is important to strengthen this part making it underlined by the current body of knowledge

"WIFI was used as the communication protocol." please reword since at this point the statement is semantically inaccurate.

"The first part presents real time information" considering that there are data ranges, more accurate would be "nearly real time"

"Air Quality Index" determination of such should be explained.

"shows an hourly aggregated noise level comparison of three monitors for the first work week of January 2021" how the data can be aggregated if the data on several months are analysed? is this configurable? In another direction, what is the shortest period of time to discover? - is it possible to get an insight on a specified hour / minute? 

In Section 3.4. although for other Sections this also would be beneficial to put direct links to the mentioned artifact. This is also the case for " obtained from the Office of Legal Consult of the executive branch of the Dominican Republic web page." and similar cases.

Section devoted to Limitations and Discussion in the context of the current body of knowledge would be beneficial. Also, it is needed to revisit existing alternative solutions and provide a brief discussion on both. 

What are the future works? This should also be added as a section. Is it planned to implement sending notifications to the users, when the level of air or other parameter pose risk to them? 

Considering that the authors several times mention the term "trust", how it was / will be measured? This should be made more clear by actually assessing it with real citizens.

It would be also beneficial to invite a native speaker, who would allow to improve the language. While the manuscript is readable, both the style, grammar etc. could be improved.

Otherwise, I find the topic very timely and the solution beneficial at the national level, also contributing to the research as a call to take an action. I believe that after the changes the study will be ready to be published and affect the world.

Reviewer 3 Report

The topic of the article is interesting, presents a good topic for readers of this Journal, but I would like to make some observations as follows:

The abstract must be improved. Must have the detailed objectives of this study and indicate which are the cities of study.

The innovation of the article should be presented to the introduction.

The article must present a brief description and location of the cities in the Dominican Republic that are the object of the study. 

Visual quality of Figures 2, 3, 13 is not good. It should be improved so that the information can be read. 

Figures 5, 6, 7, 8, 10 – They present information written in Spanish (for example: “indice de calidad del aire”)- It must be written in English.

4. Results and Discussion 

Case 1: Does the phenomenon (dust landscapes from the Sahara) only happen once a year? It would be an asset to compare the air quality index of more events/years. For example, with 2021 and 2022

4.2 Case 2: COVID-19. In the title of this subchapter, the city under study must be indicated.

Table 2 is not a platform result, so it should not be included in this chapter.

Line 383 – It is referred “Similarly in the city of Santiago de Los Caballeros, the data suggests that a more relaxed curfew increases the level of pollution, reducing the air quality.” Clarify how many (two or three?) and which cities were the object of the study.

Discussion of results needs to be improved.This chapter has to discussion the results of the study in relation to the objectives

5. Conclusions: This chapter must be improved. The conclusions must be thoroughly supported by the results presented in the article or referenced in secondary literature.

Round 2

Reviewer 1 Report

I appreciate the new revision. The authors met the points I raised and the quality is overall appropriate for publication.

Author Response

I appreciate the new revision. The authors met the points I raised and the quality is overall appropriate for publication.

We appreciate your comments as they help us to improve the study.

Reviewer 2 Report

The authors made a round of revisions. Mostly I am satisfied with the answers provided, although it was difficult to identify them, since the authors do not follow the requirements, according to which all changes should be visible.

The only comment now is the need of elaborating on Limitations. Although the authors added the section for Limitations and Future Works, mostly the Future works are discussed, while limitations are missing, especially in the light of their presence in solutions of such type, i.e. which are expected to be actively utilized by the very diverse end-users with different needs, capacities, digital skills, expectations, levels of trust and understanding of usability etc.. The authors are therefore asked to consider what are the limitations and elaborate on them.

Otherwise, the paper is improved and I believe it can become of interest for readership.

Author Response

The authors made a round of revisions. Mostly I am satisfied with the answers provided, although it was difficult to identify them, since the authors do not follow the requirements, according to which all changes should be visible.

We apologies for not activating the track change

The only comment now is the need of elaborating on Limitations. Although the authors added the section for Limitations and Future Works, mostly the Future works are discussed, while limitations are missing, especially in the light of their presence in solutions of such type, i.e. which are expected to be actively utilized by the very diverse end-users with different needs, capacities, digital skills, expectations, levels of trust and understanding of usability etc.. The authors are therefore asked to consider what are the limitations and elaborate on them.

Section 5.1 was revised considering these suggestions.The limitations of the study are clarified and future work to improve each limitation was added.

Otherwise, the paper is improved and I believe it can become of interest for readership.

Reviewer 3 Report

The article has improved and the review has met most of the suggestions.

Author Response

The article has improved and the review has met most of the suggestions.

We appreciate your comments as they help us to improve the study.